# Gold Nanoparticles Induced Size Dependent Cytotoxicity on Human Alveolar Adenocarcinoma Cells by Inhibiting the Ubiquitin Proteasome System

**DOI:** 10.3390/pharmaceutics15020432

**Published:** 2023-01-28

**Authors:** Bashiru Ibrahim, Taiwo Hassan Akere, Swaroop Chakraborty, Eugenia Valsami-Jones, Hanene Ali-Boucetta

**Affiliations:** 1Nanomedicine, Drug Delivery & Nanotoxicology (NDDN) Laboratory, School of Pharmacy, College of Medical and Dental Sciences, University of Birmingham, Birmingham B15 2TT, UK; 2School of Geography, Earth and Environmental Sciences, College of Life and Environmental Sciences, University of Birmingham, Birmingham B15 2TT, UK

**Keywords:** proteasome, deubiquitinases, AuNPs, A549 cells, cytotoxicity

## Abstract

Gold nanoparticles (AuNPs) are widely used in biomedicine due to their remarkable therapeutic applications. However, little is known about their cytotoxic effects on the ubiquitin proteasome system (UPS). Herein, the cytotoxicity of different sizes of AuNPs (5, 10, and 80 nm) on the UPS was investigated with a particular focus on deubiquitinating enzymes (DUBs) such as ubiquitin-specific proteases (USP) and ubiquitin carboxyl-terminal hydrolases (UCHL-1) in human alveolar epithelial adenocarcinoma (A549). It was found that all sizes of AuNPs reduced the percentage of viable A549 cells and increased lactate dehydrogenase (LDH) release, measured using the MTT and LDH assays, respectively. Furthermore, the 5 nm AuNPs were found to exhibit greater cytotoxicity than the 10 and 80 nm AuNPs. In addition, apoptosis and necrosis were activated through reactive oxygen species (ROS) generation due to AuNPs exposure. The internalisation of AuNPs in A549 cells increased with increasing particle size (80 > 10 > 5 nm). Interestingly, the expression of USP7, USP8, USP10, and UCHL-1 was significantly (*p* < 0.001) downregulated upon treatment with 5–30 µg/mL of all the AuNPs sizes compared to control cells. Moreover, the inhibition of these proteins triggered mitochondrial-related apoptosis through the upregulation of poly (ADP-ribose) polymerase (PARP), caspase-3, and caspase-9. Collectively, these results indicate that AuNPs suppress the proliferation of A549 cells and can potentially be used as novel inhibitors of the proteasome.

## 1. Introduction

In recent years there has been an increase in AuNPs-related research due to their potential applications in cancer treatment [1,2]. Their unique size-related (1–100 nm) physicochemical properties and high surface area to volume ratio enhance their reactivity with other molecules and enables them to cross biological barriers [3,4]. Additionally, AuNPs can be used as drug carriers to improve drug delivery and reduce side effects through more effective targeting [5,6]. Recent advances in AuNPs development have shown that they may also be ideal candidates for viral targeting and diagnosis [7,8,9] and may be used as stand-alone entities or carriers for other drugs or vaccine delivery [10,11,12].

Despite having various applications in cancer research, bioimaging, vaccines, and drug delivery, there is a growing concern about their short- and long-term health effects [13,14]. Studies have revealed that the accumulation of AuNPs may impede the biological performance of the cells through the induction of oxidative stress, reduction in metabolic activities, decrease in mitochondrial membrane potential, depletion of glutathione, and increase in lipid peroxidation, consequently leading to apoptosis and necrosis [15,16,17]. For instance, exposure to AuNPs induced the apoptosis of A549 cells and was characterised by cell shrinkage, reactive oxygen species (ROS) generation, and an increase in apoptotic protein expression [18,19]. Apart from cytotoxicity, the induction of cell cycle arrest in different cell phases upon exposure to AuNPs has also been reported [20,21]. Prasad et al. [22] found that AuNPs induced oxidative stress and G0/G1 cell cycle arrest by blocking the S phase, therefore inducing apoptotic cell death. Similar results were also reported using modified AuNPs [23]. The toxicity of AuNPs is considered to depend on their size, shape, surface charge, composition, concentrations, and cell type [24,25,26].

The UPS is the major intracellular pathway for the degradation of 80–90% of proteins in eukaryotic cells [27,28]. During the process, ubiquitin (76 amino acids) covalently binds to the proteins to be degraded in the proteasome. This is accomplished by three enzymes: ubiquitin-activating enzymes (E1), ubiquitin-conjugating enzymes (E2), and ubiquitin ligases (E3) (Figure 1) [29,30]. DUBs reverse the degradation process by removing the covalently attached ubiquitin from proteins, allowing the recycling of ubiquitin for further use by the proteasome [31]. The enzymes play an important role in controlling the process and are involved in many signalling activations such as DNA repair, apoptosis, and cell cycle progression [32,33]. Due to their regulatory functions, tumour cells have been demonstrated to depend on DUBs for proliferation [34,35]. Recently, many molecules have been tested for their toxicity against DUB activity and implicated in cancer treatment. For instance, cisplatin has been shown to suppress the proliferation of ovarian cancer through proteasome inhibition [36]. In breast cancer, the small molecule bis-benzylidine piperidone 15 (b-AP15) and platinum pyrithione (PtPT) were tested against elevated USP14, UCHL-5, and RPN11 proteases. These compounds were found to induce cell cycle arrest and cell death associated with caspase activation, endoplasmic reticulum stress, and downregulation of the endoplasmic reticulum by downregulating DUBs [37]. Although these compounds have shown a promising effect, there is still the issue of their limited efficacy, specificity, and biocompatibility [38]. Therefore, alternative DUB inhibitors are urgently needed in cancer treatment to improve efficacy and biocompatibility. Herein, we first investigated whether different sizes of AuNPs (5, 10, and 80 nm) coated with citrate can affect the UPS, with a particular focus on DUBs such as the ubiquitin-specific protease (USP) and ubiquitin carboxyl-terminal hydrolase L1 (UCHL-1) in A549 cells. Extensive insight into the underlying mechanisms of AuNPs toxicity in A549 cells was further investigated by assessing their metabolic activity, ROS, expression of apoptotic-related proteins, and their cellular uptake mechanisms.

## 2. Materials and Methods

### 2.1. Materials and Reagents

AuNPs of sizes 5, 10, and 80 nm coated with citrate were used in this study. They were all purchased from NanoComposix (San Diego, CA, USA). Phosphate-buffered saline (PBS), F12 nutrimix media, penicillin/streptomycin, 0.25% trypsin-EDTA, fetal bovine serum (FBS), and RPMI 1640 were purchased from Thermo Fisher Scientific (Paisley, UK) and the CytoTox 96^®^ non-radioactive cytotoxicity assay was from Promega (Southampton, UK). Chlorpromazine hydrochloride, wortmannin, nystatin, simvastatin, methyl-β-cyclodextrin (MβCD), menadione, staurosporine, dimethyl sulfoxide (DMSO), tris-base, glycine, and sodium chloride were purchased from Sigma Aldrich (Dorset, UK). Bovine serum albumin (BSA), acrylamide/bis-acrylamide, 4% paraformaldehyde, and ponceau were bought from Alfa Aesar (Lancashire, UK). Alexa fluor 488^®^ Annexin V/PI cell death and CellRox deep red reagent kits were purchased from Thermo Fisher Scientific (Paisley, UK). The PVDF membrane and ECL ™ were bought from GE HealthCare (Buckinghamshire, UK). USP7, USP8, USP10, UCHL-1, caspase-3, caspase-9, and PARP primary antibodies were purchased from Cell Signaling Technology (CST) (London, UK).

### 2.2. Physicochemical Characterisation of Different Sizes of AuNPs

The sizes, polydispersity index (PDI), and zeta potentials of 5, 10, and 80 nm AuNPs coated with citrate at a concentration of 10 µg/mL were measured in Ultrapure Water (UPW), and cell culture media (CCM) supplemented with 10% FBS and 1% penicillin/streptomycin at 0 and 24 h by dynamic light scattering (DLS) (Malvern Instruments Ltd., Worcestershire, UK) at 25 °C. A total of 1 mL of the dispersed AuNPs were pipetted into a disposable polystyrene cuvette (Bohemia, NC, USA) for size and PDI measurement. For zeta potential, the dispersions were pipetted into a capillary cell cuvette (Malvern Instruments Ltd., Worcestershire, UK). The morphology and size distribution of the nanoparticles was studied using a transmission electron microscope (TEM) (JEOL Ltd. Hertfordshire, UK) operating at an accelerating voltage of 80 keV (JEOL 1400 TEM, University of Birmingham, Centre for Electron Microscopy). A freshly prepared 10 µL of AuNPs (10 µg/mL) dispersed in UPW was deposited onto a copper 300 Mesh TEM grid (EM Resolutions Ltd., Sheffield, UK) and allowed to dry for 2 h in a safety cabinet before observation by TEM.

### 2.3. Cell Culture of A549 Cells

A549 cells cryopreserved at a very low passage number of five in our laboratory were thawed and cultured in F-12 nutrimix media. The media was supplemented with 10% FBS, 1% 100 µg/mL penicillin, and 100 mg/mL streptomycin at 37 °C in an incubator with a humidified atmosphere of 5% CO_2_. A549 cells were used in this study because the lungs represent the first portal to the body once nanomaterials are inhaled and therefore can be used for studying the AuNPs’ cytotoxicity on one type of lung cells.

### 2.4. Cell Viability Assays of Different Sizes of AuNPs

#### 2.4.1. MTT Assay

The viability of A549 cell lines exposed to 5, 10, and 80 nm was measured using the MTT assay. The A549 cells were seeded in flat bottom 96 well plates at a density of 7000 cells/well, diluted in 100 µL of CCM, and allowed to attach overnight at 37 °C in a 5% CO_2_ humidified incubator. The cells were treated with different concentrations of AuNPs (5–40 µg/mL) diluted in CCM for 24 and 48 h. Control cells were incubated with CCM while the positive control cells were treated with 20% DMSO diluted in CCM. The MTT solution was prepared as 5 mg/mL stock in sterile PBS and filtered with 0.22 µM Acrodisc syringe filter (Sigma Aldrich, Dorset, UK) in a microbiological safety cabinet. After the treatment, the media was removed and 120 µL of MTT solution was added, diluted in CCM at a ratio of 1:6 for 4 h. The MTT solution was replaced by adding 200 µL of DMSO for 10–15 min to solubilise the formazan crystals in the dark with gentle agitation of the shaker. The absorbance was recorded using a FLUOstar Omega microplate reader (BMG LABTECH Ltd., Aylesbury, UK) at 570 nm emission wavelength to determine the percentage cell viability of the A549 cells. The percentage cell viability was calculated using the following equation:% Cell Viability = [Absorbance of treated cells/Average absorbance of control cells] × 100

#### 2.4.2. LDH Release Assay

The LDH assay was used to measure the percentage of cell membrane damage induced by the nanoparticles associated with the release of the LDH enzyme from the cytoplasm [39]. The LDH assay was carried out using Cytotox 96 non-radioactive cytotoxicity assay Kit (Promega, Southampton, UK) according to the manufacturer’s instructions. After incubation with different sizes of AuNPs as described in the MTT assay, 50 µL of aliquots from all treated and control wells were transferred to a new flat-bottom 96-well plate and 50 µL of the reconstituted substrate was added to each well and incubated for 15 min in the dark. After the incubation period, the reaction was terminated by adding 50 µL of stop solution. The absorbance was measured using a FLUOstar Omega microplate reader (BMG LABTECH Ltd., Aylesbury, UK) at 490 nm emission wavelength. The absorbance of supernatant corresponds to the LDH released in the media which indicates the number of damaged cells caused by the AuNPs exposure. The percentage of LDH released was calculated by the following equation:% LDH released = [Absorbance of treated cells/Average absorbance of control cells] × 100

### 2.5. Cellular Uptake of Different Sizes of AuNPs

#### 2.5.1. Inductively Coupled Plasma Mass Spectrometry

ICP-MS was used to measure the amount of Au uptake by A549 cells. A549 cells were seeded in six well plates at a density of 2 × 10^5^ cells/well in 3 mL of CCM supplemented with 10% FBS and 1% penicillin/streptomycin at 37 °C. At 24 h post-seeding, the A549 cells were exposed to 5, 15, and 30 µg/mL of different sizes of AuNPs dispersed in CCM for 24 h. After the exposure, the cells were washed three times with PBS, harvested, and transferred to a glass vial (Ampulla, Lancashire, UK). The cells were digested with 3 mL of aqua regia prepared with 37% hydrochloric acid (HCl) and 70% nitric acid (HNO_3_), then 1 mL of hydrogen peroxide was added to the cell suspension. The glass vials were properly closed and heated at 120 °C for 30 min. The samples were diluted to a final acid concentration of 2% HNO_3_ and filtered with a 0.22 µM syringe filter (Sigma Aldrich, UK) to ensure the removal of any debris. The total Au concentration in the samples was measured by ICP-MS (NexION 300X, Perkin Elmer, Waltham, MA, USA) in the Helium KED mode (Helium Gas Flow at 5 mL/min). A calibration curve of gold standard series was also prepared (0, 0.25, 0.5, 1, 2, 5, 10, 20, 50, and 100 ppb) in 2% HNO_3_ to set up the ICP-MS.

#### 2.5.2. Flow Cytometry

For the measurement of AuNPs uptake using flow cytometry, A549 cells were seeded in six well plates at a density of 2 × 10^5^ cells/well in 3 mL of CCM and allowed to adhere overnight at 37 °C in 5% CO_2_. Then, the cells were exposed to different sizes of AuNPs at concentrations of 5, 15, and 30 µg/mL while the control cells were treated with CCM for 24 h. After the exposure time, the cells were washed with PBS three times to remove any unattached AuNPs and were then harvested and centrifuged for 5 min at 1500 rpm at 4 °C. The supernatant was discarded, and the cell pellet was resuspended in 500 µL of PBS. The samples were then analysed using BD LSRFortessa™ X-20 flow cytometry (BD Biosciences, Franklin Lakes, NJ, USA)) by acquiring at least 10,000 events.

#### 2.5.3. Transmission Electron Microscopy

A549 cells were seeded at a density of 2 × 10^5^ cells/well onto a 13 mm diameter sterilised coverslip in six well plates and allowed to adhere overnight at 37 °C in 5% CO_2_. The next day, the cells were incubated with different sizes of AuNPs at 30 µg/mL for 24 h. At the end of the exposure time, the cells were rinsed with PBS to remove unattached AuNPs. This was followed by fixation with 2.5% glutaraldehyde in 0.1M PO_4_ buffer diluted in PBS for 24 h at 4 °C. The samples were then taken to the Electron microscopy centre (University of Birmingham, Birmingham, UK) for processing. The cells were dehydrated in ethanol and embedded in epoxy resin. An ultrathin section (70–90 nm) parallel to the cover glass was cut with a diamond knife and mounted onto 300 mesh copper grids on a carbon film. Images were visualised with TEM (JEOL 1400) operated at an acceleration voltage of 80 kV in imaging mode.

### 2.6. Cellular Uptake Mechanisms of Different Sizes of AuNPs using Pharmacological Inhibitors

#### 2.6.1. Inductively Coupled Plasma Mass Spectrometry

A549 cells were seeded in six well plates at a density of 2 × 10^5^ cells/well in 3 mL of CCM, supplemented with 10% FBS and 1% penicillin/streptomycin at 37 °C. At 24 h post-seeding, the cells were pre-treated with either 10 µg/mL of chlorpromazine (clathrin-coated pit inhibitor), 5 µg/mL of nystatin (caveolae-mediated inhibitor), 2.5 µM of methyl-β-cyclodextrin (MβCD) (caveolae-mediated/cholesterol inhibitor), and 10 µg/mL of wortmannin (macropinocytosis inhibitor) for 1 h. This was followed by treatment with 30 µg/mL of different sizes of AuNPs dispersed in CCM for 3 h. After exposure, the cells were washed three times with PBS, harvested, and transferred to a glass vial. The cells were digested with 3 mL of aqua regia prepared with 37% HCL and 70% HNO_3_. Then, 1 mL of hydrogen peroxide was added to the cell suspension. The glass vials were properly closed and heated at 120 °C for 30 min. The samples were diluted to a final acid concentration of 2% HNO_3_ and filtered with a 0.22 µM syringe filter (Sigma Aldrich, UK) to ensure the removal of any debris. The total Au concentration in the samples was measured using the same procedure as described in Section 2.5.1.

#### 2.6.2. Flow Cytometry

The A549 cells were seeded in six well plates at a density of 2 × 10^5^ cells/well in 3 mL of CCM and allowed to adhere overnight at 37 °C in 5% CO_2_. The next day, the A549 cells were pre-treated with either chlorpromazine (10 µg/mL), nystatin (5 µg/mL), MβCD (2.5 µM), or wortmannin (10 µg/mL) for 1 h, followed by treatment with 30 µg/mL of different sizes of AuNPs dispersed in CCM for 3 h. At the end of the exposure time, the cells were rinsed with PBS to remove unattached AuNPs, followed by harvesting and centrifugation at 1500 rpm for 5 min at 4°C. The supernatant was discarded and resuspended in 500 µL of PBS. The samples were then analysed using the same procedure as described in Section 2.5.2.

### 2.7. Detection of Reactive Oxygen Species Generation

The CellRox™ deep red reagent kit was used for oxidative stress detection in A549 cells as per the manufacturer’s instructions. A549 cells were seeded in 12 well plates at a density of 2 × 10^4^ cells/well and incubated overnight to adhere. The cells were then treated with 5, 15, and 30 µg/mL of different sizes of AuNPs for 24 h while a 50 µM menadione treatment for 1 h was used as the positive control. Following the incubation, the cells were detached by trypsinisation and centrifuged at 1500 rpm for 5 min. The cells were then stained with 5 µM CellRox deep red reagent diluted in phenol-free RPMI 1640 media for 1 h at 37 °C and protected from light. The cells were then centrifuged for 5 min at 1500 rpm after the staining and the pellet was resuspended with 1 mL of RPMI 1640 media. The amount of ROS generated was then analysed immediately by BD LSRFortessa™ X-20 flow cytometry (BD Biosciences, Franklin Lakes, NJ, USA) by acquiring at least 10,000 events.

### 2.8. Annexin-V/PI Apoptosis Assay

Apoptosis induced after exposure to different sizes of AuNPs was evaluated using the Annexin V-FITC/PI assay kit (Thermo Fisher Scientific, Paisley, UK) according to the manufacturer’s instructions. A549 cells were seeded in six well plates overnight at a density of 2 × 10^5^ cells/well and treated with different concentrations of AuNPs (5, 15, and 30 µg/mL) for 24 h. The positive controls were incubated with either 10% DMSO or 1 µM Staurosporine for 6 h as inducers of necrosis and apoptosis, respectively. The cells were then collected by trypsinisation and centrifuged at 1500 rpm for 5 min at 4 °C. The pellet was resuspended with 100 µL of Annexin binding buffer and incubated with 5 µL of Annexin V-FITC and 1 µL of PI for 15 min in the dark at room temperature. After the incubation, the cell suspension was further diluted with 400 µL of Annexin binding buffer and analysed by BD LSRFortessa™ X-20 flow cytometry (BD Biosciences, Franklin Lakes, NJ, USA) by acquiring at least 10,000 events. Electronic compensation was performed between the fluorophores to prevent any spectral overlap. The maximum excitation and emission wavelengths for the Alexa Fluor^®^ 488 Annexin V stain were 488/499 nm, and 535/617 nm for the PI stain. The recorded data were analysed by FlowJo software version 10.7.1 (Ashland, Oregon, USA) and expressed as the percentage cell population stained with PI or Annexin V.

### 2.9. Western Blotting

For Western blotting, A549 cells were seeded at a density of 2.5 × 10^5^ cells/well in six well plates and incubated for 24 h to adhere to the plates. Cells were treated with different sizes of AuNPs at concentrations of 5, 15, and 30 µg/mL for 24 h with 10 µg/mL MG132 proteasome inhibitor being used as a positive control. The cells were then lysed with an NP40 lysis buffer containing a protease inhibitor cocktail (Thermo Fisher Scientific, UK) on ice. The cells were then collected in a fresh Eppendorf tube and centrifuged at 13,000 rpm for 10 min at 4 °C. Protein concentrations were measured using the Bradford reagent. Afterwards, 20 µg of the lysates were separated using sodium dodecyl sulfate-polyacrylamide gel electrophoresis (SDS-PAGE) (Bio-Rad Ltd., Watford, UK) and transferred to a PVDF membrane. The membranes were blocked in TBST (Tris Base, NaCl, pH 8.3) prepared with Tween-20 (0.1%) and skim milk (5%) for 1 h at room temperature with gentle agitation. The membranes were then washed three times with TBST for 8 min, followed by incubation with primary antibodies diluted in TBST containing 5% bovine serum albumin (1:1000) overnight at 4 °C on the shaker. After washing with TBST, the membranes were then incubated with a secondary antibody (horseradish peroxidase-conjugated anti-rabbit or anti-mouse IgG) diluted 1:2000 in TBST containing 5% BSA for 1 h at room temperature. Subsequently, the membranes were washed with TBST and enhanced chemiluminescent reagents were added to detect the signal intensity of the proteins using G:BOX Chemi XX6/XX9 (Syngene, Cambridge, UK).

### 2.10. Statistical Analysis

Statistical analysis was performed using GraphPad Prism software version 9.0. BD LSR-Fortessa X20 flow cytometry data were analysed by FlowJo software version 10.7.1 (Ashland, OR, USA). All the data were presented as the mean ± standard deviation for at least three independent experiments. The one-way analysis of variance (ANOVA) was used to calculate the statistical significance between the control and AuNPs treated groups, followed by Bonferroni post hoc multiples comparison.

## 3. Results

### 3.1. Physicochemical Characterisation of Different Sizes of AuNPs

The hydrodynamic size distribution, polydispersity index (PDI), and zeta potential of 5, 10, and 80 nm AuNPs used in this study was measured using DLS in UPW and CCM at 0 and 24 h. The DLS measurement revealed that the hydrodynamic diameter of 5, 10, and 80 nm were 12.92 ± 1.45 nm, 16.40 ± 0.39 nm, and 85.85 ± 0.76 nm, respectively, in UPW at 0 h (Figure 1A). Moreover, when the AuNPs were kept in UPW for 24 h, a slight increase in hydrodynamic diameter was observed compared to the hydrodynamic diameter at 0 h, as shown in Figure 1A. In addition, the PDI also showed an apparent change in AuNPs stability for 0 and 24 h in the UPW. For instance, the 5 nm AuNPs showed an initial PDI value of 0.38 ± 0.08 for 0 h which changed slightly to 0.48 ± 0.09 after 24 h. Likewise, the PDI values of 10 nm AuNPs increased from 0.27 ± 0.01 at 0 h to 0.36 ± 0.02 after 24 h, whereas 80 nm showed a noticeable PDI decrease at 24 h. To further understand how AuNPs behave in the exposed CCM, we suspended the particle in CCM containing 10% FBS and 1% penicillin/streptomycin at the concentration of 10 µg/mL. As seen in Figure 1B, there was an increase in the hydrodynamic diameter of all the AuNPs once exposed to CCM when compared to the UPW. This could be due to the presence of macromolecules in the CCM media such as vitamins, amino acids, antibiotics, glucose, and protein supplements. The PDI values in the CCM show higher values compared to the UPW with all the nanoparticles, slightly fluctuating between 0 and 24 h. While the PDI values slightly increased for the 5 and 10 nm AuNPs after 24 h, there was a slight decrease in PDI value for the 80 nm AuNPs as shown in Figure 1B. As expected, the measured zeta potential showed that all the nanoparticles were negatively charged in UPW and CCM (Figure 1C–D). Interestingly, the zeta potential values of the AuNPs showed an apparent decrease in UPW for all nanoparticles as shown in Figure 1C. However, the zeta potential of the AuNPs noticeably increased when assessed in CCM at both time points (Figure 1D). TEM images of all three sized AuNPs showed that the particles are spherical with a uniform size distribution (Appendix A).

### 3.2. AuNPs Decreased Cell Viability and Increased LDH Release in A549

In the current study, the cytotoxicity of 5, 10, and 80 nm AuNPs was measured based on the metabolic activity and LDH release from A549 cells using the MTT and LDH assays, respectively. As shown in Figure 2, the decrease in percentage cell viability and release of LDH is related to the increase in AuNPs concentrations (5–40 µg/mL) and incubation period (24–48 h). In Figure 2A, the cell viability of A549 cells was maintained with up to 10 µg/mL for 5, 10, and 80 nm AuNPs after 24h incubation. As the concentrations increased above 10 µg/mL, the cell viability of A549 cells decreased for all three sizes of AuNPs. AuNPs at 40 µg/mL showed the most significant decrease in cell viability (9.24%, 12.91%, and 24.08%) for the 5, 10, and 80 nm, respectively. Using the same concentrations, the cell viability measured after 48 h dramatically decreased at a greater rate (Figure 2B), with low concentration (5 µg/mL) having a minor impact (91%, 87.1%, and 85.4%) for the 80, 10, and 5 nm, respectively. The membrane integrity effect induced by AuNPs on A549 cells was also assessed by LDH release in cell culture media. It was found that AuNPs showed a dose-dependent increase in LDH release at a concentration of 20 µg/mL and higher (Figure 2C). In addition, an increase in LDH release was observed from a concentration of 10 µg/mL (5.2% increase compared to control) for all the nanoparticles (Figure 2D). A statistically significant difference (*p* < 0.01) between the treated concentrations and control was observed at the exposure levels (25–40 µg/mL) and (20–40 µg/mL) for 24 and 48 h, respectively. In summary, the assays revealed that 5 nm was more cytotoxic than 10 and 80 nm at both 24 and 48 h. Although mass concentrations were used for the viability assays, we further explored if the cytotoxicity of the nanoparticles is related to the surface area of the AuNPs and particle number concentration by normalising the surface area and particle number concentration from the corresponding concentrations used in the MTT assay. We found that surface area varies with the variation of particle diameter; at the same concentration (40 µg/mL), 5 nm AuNPs (4.13 × 10^−5^ m^2^/g) have a higher surface area compared to the 10 nm (1.09 × 10^−5^ m^2^/g) and 80 nm (1.02 × 10^−6^ m^2^/g) AuNPs which could explain why they are more cytotoxic (Appendix A). Conversely, normalisation of mass concentration to particle number concentration revealed that 80 nm AuNPs were more cytotoxic than 5 and 10 nm AuNPs due to the fact that 80 nm AuNPs require a lower amount of particle number concentration (8.46 × 10^8^ NPs/mL) to inhibit the percentage of cell viability at the same mass concentration (40 µg/mL) compared to the 5 nm and 10 nm AuNPs (3.17 × 10^13^ NPs/mL, 2.49 × 10^12^ NPs/mL) (Appendix A).

Based on these experiments, we selected concentrations of 5, 15, and 30 µg/mL for subsequent experiments to evaluate other modes of toxicities.

### 3.3. AuNPs Uptake in A549 cells by ICP-MS, Flow Cytometry, and Transmission Electron Microscopy (TEM)

ICP-MS, flow cytometry, and TEM were used to measure the cellular uptake of different sizes of AuNPs after incubation for 24 h. When the mass concentrations of AuNPs in the cells were quantitatively measured using ICP-MS, it was observed that the uptake of AuNPs was highly dependent on the concentration and size of the nanoparticles, as shown in Figure 3A. The uptake was lowest at 5 µg/mL for the 5, 10, and 80 nm AuNPs (with 5.62, 22.90, and 38.97 Au pg/10^4^ cells, respectively). As the concentration was tripled, the uptake slightly increased for the 10 nm (25.84 Au pg/10^4^ cells) and 80 nm (59.08 Au pg/10^4^ cells) but did not increase for the 5 nm AuNPs. Interestingly, when the concentration of AuNPs was further increased to 30 µg/mL, the uptake only increased with the 80 nm nanoparticles (118.38 Au pg/10^4^ cells). It was interesting to find that the order of AuNPs uptake (80 > 10 > 5 nm) did not correspond with the order of cell viability decrease and LDH release, thus, the 80 nm AuNPs proved to be less cytotoxic than the 5 and 10 nm.

We then used another method to assess the cellular uptake of AuNPs into A549 cells. Flow cytometry was used to measure the intensity signal of side scatter (SSC) which indicates the cells’ internal granularity and complexity [40]. This technique was previously used to evaluate the uptake of functionalised carbon nanotubes in A549 cells [41], and AuNPs in MDA-MB-231 cells [42]. When the cells were incubated with the same concentrations used for ICP-MS at 24 h, the SSC intensity signals increased in a dose-dependent manner when compared with the control, which could be attributed to the uptake of AuNPs (Figure 3B). On the other hand, a maximal SSC intensity was observed with the 80 nm size, suggesting greater uptake compared to the 5 and 10 nm (Appendix A). This is in complete agreement with that of ICP-MS which also showed greater uptake of the 80 nm. In order to accurately assess the localisation of AuNPs in the cells during the uptake process, A549 cells were incubated with 30 µg/mL of different sizes of AuNPs for 24 h and sectioned using TEM. Ultra-thin sections inside the cells were visualised with TEM to assess the localisation of AuNPs in various organelles. All AuNPs were found to be localised within vesicles of cells as shown in Figure 3C–E. Moreover, the 5 nm AuNPs were also seen to be slightly bound to the cell surface (Figure 3C).

### 3.4. Assessment of the Uptake Mechanism(s) of Different Sizes of AuNPs in A549 Cells

Chlorpromazine, nystatin, MβCD, and wortmannin endocytic inhibitors were selected to investigate the uptake mechanism(s) of AuNPs in A549 cells. These pharmacological inhibitors were chosen because they have been previously used to inhibit specific endocytotic uptake mechanisms through the depletion of cholesterol in the biological membrane (MβCD and nystatin), serum protein transferrin inhibition blocking of transglutaminase 2 enzyme (chlorpromazine), and show fewer effects on the actin cytoskeleton (wortmannin) [43,44,45,46]. First, we assessed the toxicity of these inhibitors in A549 cells using the MTT assay after 1 h of incubation (Appendix A). We selected a concentration that is non-toxic to the cell membrane to avoid interference with the actin filaments. As shown in Figure 4A, the A549 cells were pre-treated with chlorpromazine (10 µg/mL) for 1 h. Chlorpromazine is used to inhibit clathrin-coated pit formation in order to assess whether it will disturb the internalisation of AuNPs. As assessed by ICP-MS, the pre-incubation with chlorpromazine demonstrated a stronger inhibition of the uptake for 5 nm (11.34 Au pg/10^4^ cells) and 10 nm (11.85 Au pg/10^4^ cells) compared to 80 nm (32 Au pg/10^4^ cells) in A549 cells. This indicates that all the AuNPs could be internalised via clathrin-mediated endocytosis. In the presence of nystatin, which specifically inhibits caveolae-medicated endocytosis, the internalisation of all the AuNPs remained relatively close (11.12, 11.71, and 11.17 Au pg/10^4^ cells) for 5, 10, and 80 nm respectively. A similar trend was also observed when the cells were incubated with another caveolae inhibitor or cholesterol inhibitor (MβCD). Likewise, when cells were pre-treated with a specific macropinocytosis inhibitor (wortmannin), the uptake of 5 and 10 nm AuNPs decreased (9.63 and 13.29 Au pg/10^4^ cells, respectively) compared to the control (100 Au pg/10^4^ cells). However, the uptake of 80 nm reduced to 23.41 Au pg/10^4^ cells but was still much higher than that of the 5 and 10 nm AuNPs. Similar outcomes were observed when A549 cells were pre-incubated with the same endocytic inhibitors and the change in SSC signal intensity was assessed using flow cytometry as shown in Figure 4B. A dot plot analysis of the SSC against FSC signal intensities can be found in Appendix A. Herein, we show that the uptake of all AuNPs could be either via clathrin- or caveolae-mediated endocytosis.

### 3.5. Upregulation of ROS Production Caused by Different Sizes of AuNPs

Reactive oxygen species (ROS) are molecules that are formed as a by-product of the normal metabolism of oxygen [47]. They are regarded as activators of many signalling pathways in tumour cells such as AKT signalling and the apoptotic and Wnt pathways [48]. The production of these molecules is triggered mainly by exposure to environmental stress and toxic materials, which may also include nanoparticles. In our study, we used the CellRox deep red reagent kit to detect the production of ROS by different sizes of AuNPs after 24 h incubation at a concentration of 5, 15, and 30 µg/mL. In total, 50 µM of menadione was used as an inducer of oxidative stress (positive control) in A549 cells. Figure 5A–C shows that the fluorescence intensity gradually increased after exposure to various concentrations of different sizes of AuNPs which corresponds to an increase in intracellular ROS generation in A549 cells. For instance, 5 nm shows the highest increase in fluorescence intensity at 30 µg/mL compared to 10 and 80 nm AuNPs. When the mean fluorescence intensity (MFI) was quantified, 5 nm AuNPs showed a greater ROS generation in A549 cells when incubated with 5–30 µg/mL compared to the control. Notably, a lower MFI (188 ± 2.65) was observed with 10 nm AuNPs when compared with 5 nm AuNPs (211 ± 1.78) at 30 µg/mL. However, at 5 µg/mL, 10 nm AuNPs showed a significant increase in MFI (140 ± 1.53) compared to 5 nm AuNPs (127.33 ± 1.52). In contrast, 80 nm AuNPs successfully showed a dose-dependent increase in MFI from 87.40 ± 10.67 at 5 µg/mL to 178 ± 15.39 at 30 µg/mL (*p* < 0.001 compared to the control). Overall, 30 µg/mL was observed to induce a greater ROS production in all the AuNPs. In addition, 5 nm AuNPs showed significant effects compared to 10 and 80 nm AuNPs as seen in Figure 5D.

### 3.6. Mechanism of Cell Death of Different Sizes of AuNPs Assessed by Annexin-V/PI

We studied two mechanisms by which A549 cell death could be induced upon exposure to different sizes of AuNPs using flow cytometry. During early apoptosis, the cells undergo physiological changes which are characterised by the flip flop of the phosphatidyl-serine to the outer layer of the biological membrane to which fluorescently labelled Annexin V binds and emits green fluorescence signals, showing that apoptosis is occurring. At a late stage of the process, propidium iodide (PI) enters the cells, binds to the nucleus (DNA), and emits red fluorescence, indicating necrosis. As shown in Figure 6A,B, 5 and 10 nm AuNPs slightly induced both apoptosis and necrosis in a similar manner as the concentration of AuNPs increased. However, 80 nm AuNPs induced minimal apoptosis and necrosis compared to 5 and 10 nm (Figure 6C). FlowJo scatter dot plots can be found in the supporting information (Appendix A).

### 3.7. Effects of Different Sizes of AuNPs on 19S Proteasome DUBs Enzymes

To establish whether AuNPs affect the UPS, we investigated the expression of USP7, USP8, USP10, and UCHL-1 which are involved in the removal of covalently attached ubiquitin from target proteins to be degraded using Western blotting. The enzymes’ activities regulate various signalling pathways, cell homeostasis, and protein stability and are heavily linked to cancer proliferation [49]. In A549 cells, the enzymes are highly expressed [50]. Exposure to AuNPs resulted in the strong inhibition of these enzymes in a dose-dependent manner at 24 h. At the lowest concentration of AuNPs (5 µg/mL), we found that all AuNPs did not inhibit the activity of USP7, USP8, USP10, and UCHL-1 (Figure 7A). Likewise, at 15 µg/mL AuNPs slightly inhibited the activities of these enzymes compared to control cells and the lower concentration. However, exposure of A549 cells to 30 µg/mL AuNPs resulted in a significant downregulation of these proteins. Interestingly, 30 µg/mL AuNPs inhibited the expression of proteins in a similar way to the positive control (MG132) which has been shown to inhibit the proteasome. The changes of the relative protein expression of DUBs exposed to 5 nm AuNPs decreased significantly (*p* < 0.001) in a dose-dependent manner with a greater decrease observed at 30 µg/mL for USP7, USP8, USP10, and UCHL-1 as shown in Figure 7B. Similarly, 10 nm AuNPs decreased the relative protein expressions of USP7, USP8, USP10, and UCHL-1 as shown in Figure 7C (*p* < 0.001). In addition, 80 nm AuNPs significantly (*p* < 0.001) decreased the relative protein expression of USP7, USP8, USP10, and UCHL-1 at 30 µg/mL (Figure 7D).

### 3.8. Effects of Different Sizes of AuNPs on the Expression Level of Apoptosis-Related Proteins

We next assessed the induction of apoptosis of 5, 10, and 80 nm AuNPs by checking the expression of Poly (ADP-ribose) polymerase (PARP), caspase-3, and caspase-9 when incubated with 5–30 µg/mL AuNPs for 24 h. Following exposure to 5 µg/mL AuNPs, the expression of PARP and caspase-3 did not increase compared to the control. As cells were incubated with 15 µg/mL AuNPs, PARP, caspase-3, and caspase-9 were slightly upregulated as shown in Figure 8A. As the concentration increased to 30 µg/mL AuNPs, PARP, caspase-3, and caspase-9 were significantly upregulated, indicating the induction of apoptosis at higher concentrations. As shown in Figure 8B, the relative protein expression of pro-apoptotic proteins was upregulated in a dose-dependent manner when incubated with 5 nm AuNPs with the highest concentration showing a higher upregulation of PARP (1.54), caspase-3 (1.33), and caspase-9 (1.57) compared to the control (0.522) (*p* < 0.001). In addition, Figure 8C shows that the relative protein expression of PARP, caspase-3, and caspase-9 was significantly increased when treated with 30 µg/mL of AuNPs as shown in Figure 8C. Likewise, following the treatment of A549 cells with 80 nm AuNPs, the relative protein expression of PARP, caspase-3, and caspase-9 increased but to a lesser extent compared to 5 and 10 nm AuNPs (*p* < 0.001) (Figure 8D).

## 4. Discussion

AuNPs are extensively used in biomedicine because of the relative ease of controlling their physicochemical properties, (e.g. size and shape), and their apparent biocompatibility compared with other metallic nanoparticles [51,52]. However, it has also been proposed that increasing exposure to AuNPs through inhalation and ingestion can potentially cause adverse effects on human health even at a low dose [53]. In this study, we investigated the cytotoxicity of different sizes of AuNPs and their effects on the UPS of A549 cells. Initially, the hydrodynamic size of all the AuNPs was measured and was found to be slightly lower at 0 h compared to 24 h in both UPW and CCM. In addition, the size distribution of the particles was greater in CCM compared to UPW. This is because the characteristics of nanoparticles change in biological media in the presence of growth-stimulating factors such as vitamins, amino acids, and supplemented FBS. These growth stimulants could lead to the formation of a protein corona [54]. Moreover, all the AuNPs sizes exhibited negative surface charges in UPW and CCM, with a noticeable increase at 24 h, demonstrating that certain serum proteins in the media could mask the surface charge of the AuNPs [55].

We then looked at the biological effects of AuNPs on A549 cells. It was found that AuNPs induced cytotoxicity to A549 cells in an increasing order of their sizes (5 > 10 > 80 nm), with the smallest AuNPs (5 nm) causing the most cytotoxicity. In addition, when cells were exposed to a high concentration of 40 µg/mL for 24 and 48 h, the cell viability significantly decreased while the LDH release increased, indicating that cytotoxicity occurs at high AuNPs concentrations with all AuNPs sizes. In addition, these results demonstrate that the cytotoxicity of AuNPs in vitro depends not only on their sizes but also on their concentrations and incubation times, which is similar to what was previously reported [56,57]. Moreover, we observed that AuNPs affected oxidative phosphorylation in the mitochondria and caused the generation of intracellular ROS in A549 cells. The generation of intracellular ROS upon exposure to AuNPs can result in oxidative damage to mitochondrial proteins, redox imbalance, and hindrance in the ability of the cells to carry out metabolic activities. This can consequently lead to pathological conditions [58,59]. In our study, we found that ROS generation was dependent on the concentration and size of AuNPs. Surprisingly, 80 nm AuNPs were found to induce greater ROS generation than 10 nm, even though they showed the least cytotoxicity in the MTT and LDH assays. Using ICP-MS, flow cytometry, and TEM, we then investigated if the toxicity of AuNPs is related to their cellular uptake after 24 h incubation. We observed that A549 cells uptake 80 nm AuNPs to a greater extent (by mass), especially at high concentrations, compared to 10 and 5 nm, indicating that the uptake of nanoparticles increased proportionally with the increase in their size. Interestingly, this shows that the cytotoxicity of AuNPs does not correlate with their cellular uptake. The degree of uptake of AuNPs likely depends on other factors such as the diffusion of the particle, the density of the particles, and interaction with the cellular membrane [60,61,62]. Our cellular uptake results are in agreement with those of Chithrani et al. [63] who compared the uptake of different sizes of AuNPs in HeLa cells and found 50 nm to be the ideal AuNPs size for their cellular internalisation. Furthermore, the TEM imaging of A459 cells showed all nanoparticles were localised in vesicles, as evidenced by endosome formation. To gain further insight into the different mechanisms of AuNPs internalisation, we pre-incubated A549 cells with different pharmacological inhibitors for 1 h, followed by 30 µg/mL of different sizes of AuNPs (5, 10, and 80 nm) for 3 h. It was previously suggested that A549 cells can internalise AuNPs through pinocytosis [64,65]. Pinocytosis is an ideal internalisation mechanism for endothelial cells [17]. Accordingly, a particle size between 1–200 nm is expected to be internalised through either clathrin- or caveolae-mediated endocytosis [66]. In our current study, we found that the internalisation of AuNPs gradually decreased when pre-incubated with clathrin- or caveolae-mediated inhibitors. This suggests that the uptake of our AuNPs was via clathrin- and caveolae-dependent pathways.

Several cytotoxicity studies have suggested that ROS generation could be a prominent mechanism of AuNPs toxicity [67,68]. Furthermore, we have investigated the possible mechanisms of cell death, especially that the activation of apoptosis and necrosis following treatment with AuNPs can be related to their cellular uptake and ROS generation as reported by others [69,70]. As shown in our flow cytometry results using the double staining of Annexin V/propidium iodide, all the AuNPs sizes induced both apoptosis and necrosis in a dose-dependent manner. Our study suggests that the cytotoxicity of AuNPs in A549 cells may involve different molecular mechanisms and aligns with the work of Xia et al. [71] who demonstrated that 5 nm AuNPs induced both apoptosis and necrosis through the production of ROS.

The UPS plays a critical role in protein degradation, including misfolded proteins, intracellular proteins involved in cell cycle arrest, overexpressed proteins, and apoptosis-related proteins [72]. Dysregulation of UPS has been reported to be associated with the pathogenesis and progression of various tumour cells, therefore inhibition of UPS activity might be an important target in cancer treatment [33,37,73]. Although the inhibitors of proteasome such as bortezomib, carfilzomib, and MG 132 have been identified and proven to suppress the proliferation of tumour cells via proteasome inhibition [38,74,75,76], the lack of specificity and solubility of these compounds prompts an urgent call for other effective proteasome inhibitors. We have, therefore, tested the effectiveness of different sizes of AuNPs on proteasome inhibition, with a specific focus on 19S proteasome DUB activity such as USP7, USP8, USP10, and UCHL-1. The Western blotting analysis confirmed that 30 µg/mL of all sizes of AuNPs could downregulate the expression of these enzymes in the same way as the 10 µg/mL positive control (MG 132). This is the first study to show that AuNPs can affect the UPS of A549 cells. Others have shown that TiO_2_ nanoparticles downregulated the expression of UCHL-1 in rat PC12 cells [77]. We further investigated if the inhibition of 19s proteasome DUBs could lead to the induction of apoptosis since tumour cells use these proteins in the degradation of tumour suppressors [78,79]. Interestingly, it was found that AuNPs treatment in A549 cells upregulates the levels of PARP, caspase-3, and caspase-9 in a dose-dependent manner starting at 15 µg/mL of NPs. This agrees with previous findings which demonstrated that AuNPs inhibited the proliferation of cells through the induction of apoptosis [70,80,81]. For instance, the study agrees with Jawaid et al. [82] who confirmed the size-dependent apoptotic induction of different sizes of AuNPs (2, 40, and 80 nm) through caspase-3 upregulation.

## 5. Conclusions

In conclusion, the present study was undertaken to investigate the effects of different sizes of AuNPs on the UPS in A549 cells. The exposure of A549 cells to different sizes of AuNPs resulted in a decrease in percentage cell viability, an increase in LDH release, induction of apoptosis, and intracellular ROS generation that may be involved in cellular oxidative stress. A549 cells’ toxicity following treatment with AuNPs was observed to be highly dependent on their size and concentration. Furthermore, exposure to AuNPs downregulates the expression of USP enzymes such as USP7, USP8, USP10, and UCHL-1 and upregulates the expression of caspase-3, caspase-9, and PARP. Thus, this study sheds light on the mechanisms of AuNPs cytotoxicity on a lung carcinoma cell lines, as well as highlights their inhibitory effects on the UPS (Figure 1).

## Figures and Tables

**Figure 1 pharmaceutics-15-00432-f001:**
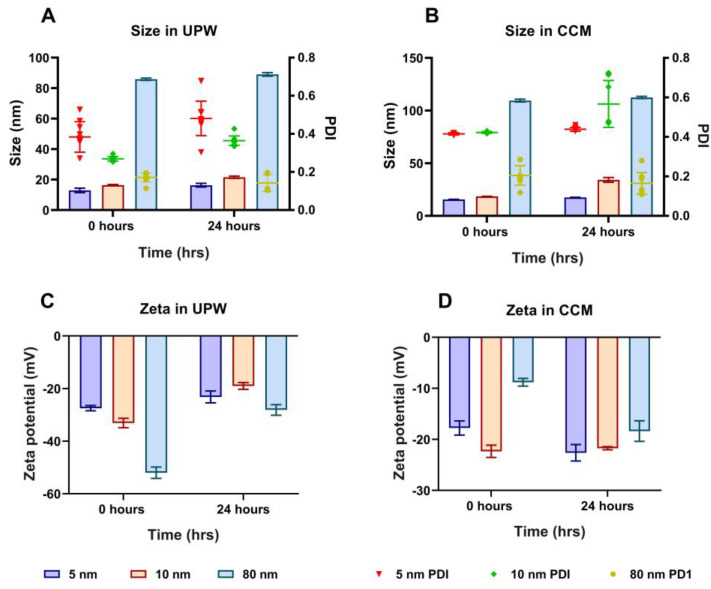
Characterisation of different sizes of AuNPs by DLS in UPW and CCM. Hydrodynamic size distribution and PDI of 5, 10, and 80 nm AuNPs in (**A**) UPW after 0 and 24 h and (**B**) CCM after 0 and 24 h. Zeta potential values of 5, 10, and 80 nm AuNPs in (**C**) UPW after 0 and 24 h and (**D**) CCM after 0 and 24 h. Plotted graphs represent the means ± standard deviation (SD) of 10 different measurements.

**Figure 2 pharmaceutics-15-00432-f002:**
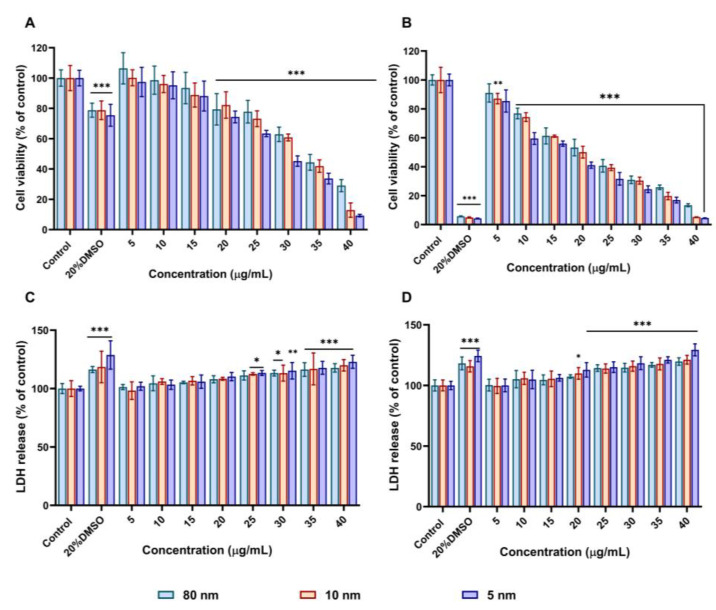
Cell viability and lactate dehydrogenase (LDH) release of A549 cells treated with 5–40 µg/mL of different sizes of AuNPs measured using MTT and LDH assays. Percentage of cell viability after incubation with AuNPs for (**A**) 24 h and (**B**) 48 h. Percentage LDH release in media after incubation with AuNPs for (**C**) 24 h and (**D**) 48 h. 20% DMSO was used as a positive control. Plotted graphs represent the means ± SD of six independent experiments. Bars with an asterisk (*) show statistical differences (* *p* < 0.05, ** *p* < 0.01, and *** *p* < 0.001) compared with the control.

**Figure 3 pharmaceutics-15-00432-f003:**
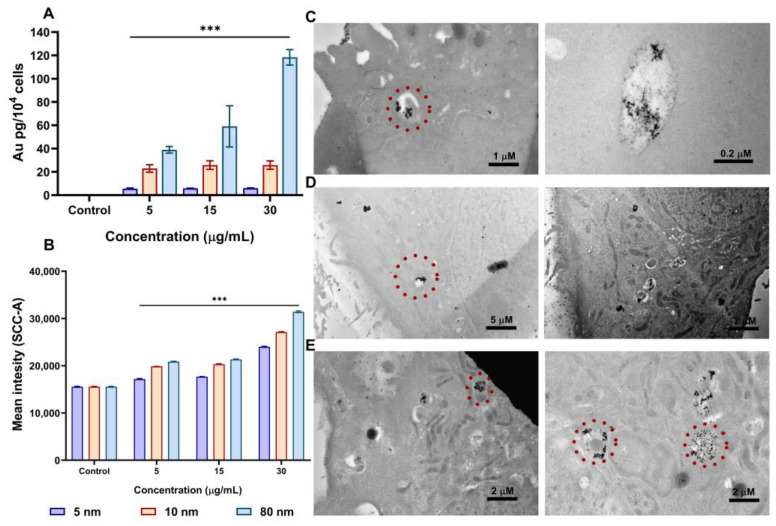
Cellular uptake of different sizes of AuNPs in A549 cells after 24 h incubation. (**A**) The Au content in A549 cells measured with inductively coupled plasma mass spectrometry (ICP-MS) (**B**) Side scatter (SSC) intensity changes due to the uptake of AuNPs analysed using flow cytometry. TEM Images of the internalisation of AuNPs into A549 cells after treatment with 30 µg/mL for 24 h of (**C**) 5 nm, (**D**) 10 nm, and (**E**) 80 nm AuNPs. AuNPs are indicated by circles. Plotted graphs represent the means ± SD of three independent experiments. Bars with an asterisk (*) show a statistical difference (*** *p* < 0.001) when compared with the control.

**Figure 4 pharmaceutics-15-00432-f004:**
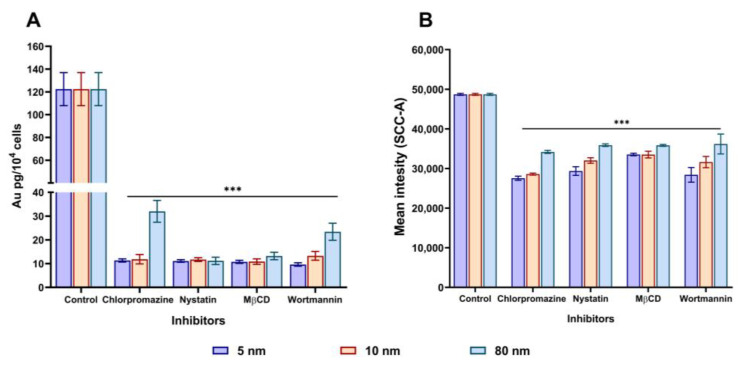
Effects of pharmacological inhibitors on the internalisation of different sizes of AuNPs (30 µg/mL) in A549 cells. Cells were separately pre-incubated with chlorpromazine, nystatin, MβCD, and wortmannin for 1 h, followed by incubation with AuNPs for 3 h and analysed by (**A**) ICP-MS and (**B**) flow cytometry. Plotted graphs represent the means ± SD of three independent experiments. Bars with an asterisk (*) show a statistical difference (*** *p* < 0.001) when compared with the control.

**Figure 5 pharmaceutics-15-00432-f005:**
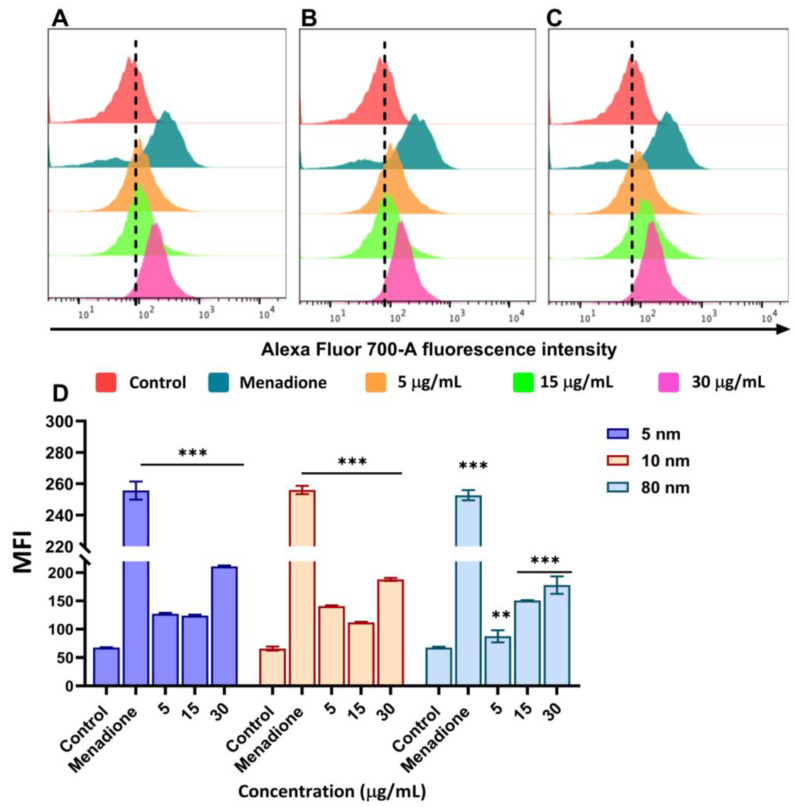
ROS production in A549 cells following exposure to different sizes of AuNPs analysed using flow cytometry with the CellRox deep red kit. Histograms of flow cytometry treated with (**A**) 5 nm (**B**) 10 nm and (**C**) 80 nm AuNPs. (**D**) Bar graphs represent the median fluorescence intensity (MFI) of A549 cells exposed to 5, 10, and 80 nm AuNPs. Menadione (50 µM) was used as a positive control. Plotted graphs represent the means ± SD of three independent experiments. Bars with an asterisk (*) show a statistical difference (** *p* < 0.01, and *** *p* < 0.001) when compared with the control.

**Figure 6 pharmaceutics-15-00432-f006:**
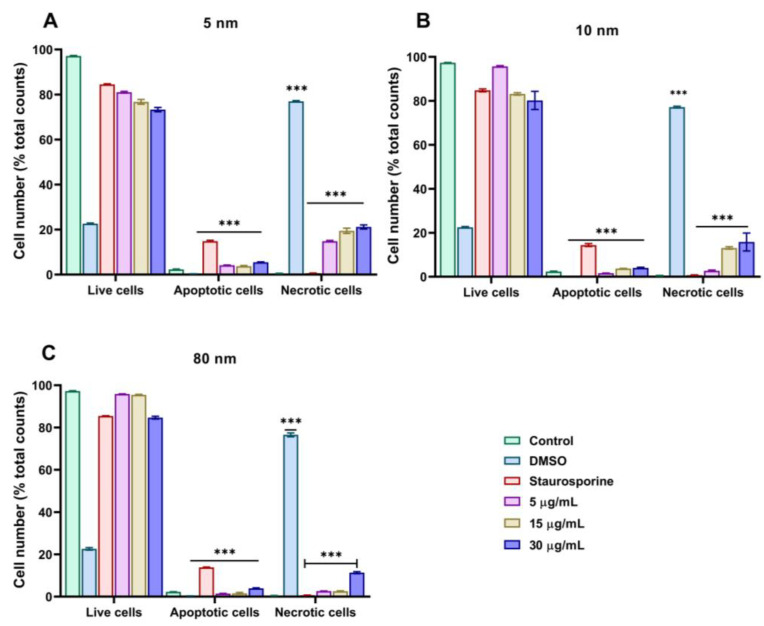
Bar graph analyses of apoptosis and necrosis assessed by flow cytometry after the incubation of A549 cells with (**A**) 5 nm, (**B**) 10 nm, and (**C**) 80 nm AuNPs at concentrations of 5–30 µg/mL. Treatment with 1 µM of staurosporine and 10% DMSO for 6 h were used as positive controls for apoptosis and necrosis, respectively. Plotted graphs represent the means ± SD of three independent experiments. Bars with an asterisk (*) show a statistical difference (*** *p* < 0.001) when compared with the control.

**Figure 7 pharmaceutics-15-00432-f007:**
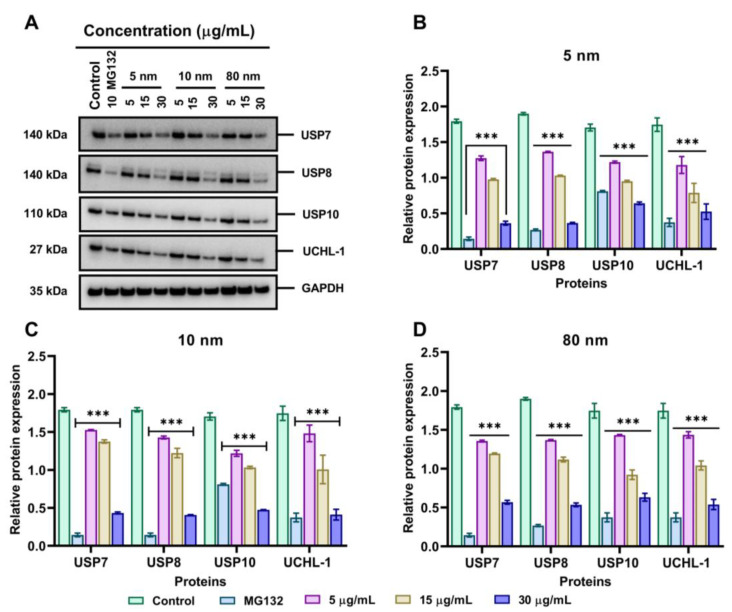
AuNPs downregulate the expression of DUBs in A549 cells. (**A**) Expression levels of DUBs detected by Western blot after A549 cells were incubated with 5, 15, and 30 µg/mL of different sizes of AuNPs for 24 h. MG132 (10 µg/mL) was used as a positive control. The relative protein expression signal intensity was semi-quantified by ImageJ software for (**B**) 5 nm, (**C**) 10 nm, and (**D**) 80 nm AuNPs. Plotted graphs represent the means ± SD of three independent experiments. Bars with an asterisk (*) show a statistical difference (*** *p* < 0.001) when compared with the control.

**Figure 8 pharmaceutics-15-00432-f008:**
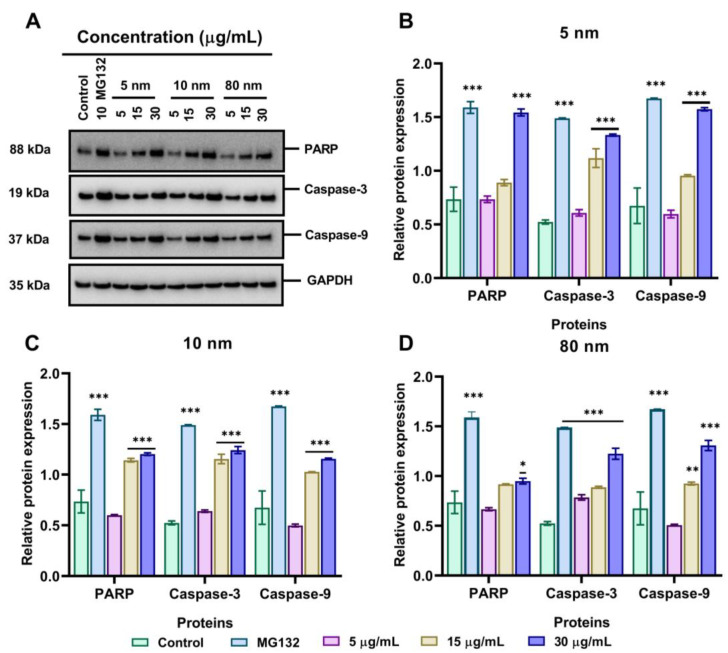
AuNPs induced apoptosis in A549 cells through a mitochondrial signalling pathway. (**A**) The expression of apoptosis-related protein levels detected by Western blot after A549 cells were incubated with 5, 15, and 30 µg/mL of different sizes of AuNPs for 24 h. MG132 (10 µg/m) was used as a positive control. The relative expression signal intensity of apoptosis-related proteins semi-quantified by ImageJ software for (**B**) 5 nm, (**C**) 10 nm, and (**D**) 80 nm AuNPs. Plotted graphs represent the means ± SD of three independent experiments. Bars with an asterisk (*) show a statistical difference (* *p* < 0.05, ** *p* < 0.01, *** *p* < 0.001) when compared with the control.

**Scheme 1 pharmaceutics-15-00432-sch001:**
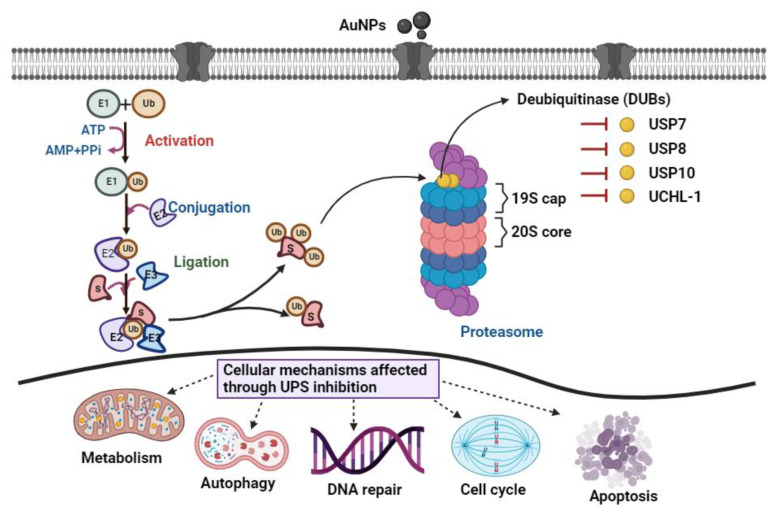
UPS diagram. Ubiquitin-activating enzymes (E1) activate the first step of the process which forms a thioester bond in the presence of a higher energy compound (ATP). The activated ubiquitin is then transferred to the ubiquitin conjugation enzyme (E2) via a transthiolation reaction. Ubiquitin ligases (E3) provide substrate specificity for the reaction and catalyse the covalent binding of ubiquitin to the target substrate through an isopeptide bond. The polyubiquitinated substrate moves to the proteasome for degradation. Finally, DUBs associated with the 19S proteasome remove conjugated ubiquitin from the substrate prior to the degradation.

## Data Availability

All the data presented here are available on request from the corresponding authors.

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
