# Peer review of "Gold Nanoparticles Induced Size Dependent Cytotoxicity on Human Alveolar Adenocarcinoma Cells by Inhibiting the Ubiquitin Proteasome System"

_pharmaceutics, 2023, doi:10.3390/pharmaceutics15020432_

Round 1
Reviewer 1 Report
Authors have conducted a systematic study on the cytotoxicity of gold nano articles with different sizes. Their results provide good reference for gold nano particles aiming for in vivo application in biomedicine field.
1. What is the main question addressed by the research? Authors tried to address the cytotoxic effects of gold nanoparticles on UPS of human aveolar epithelial adenocarcinoma, with focus on its effect on deubiquitinating enzymes. Authors were able to show the greater cytotoxic effects of Au NPs with smaller size (5 nm) despite greater uptake efficiency of larger Au NPs. Authors further revealed the regulation of DUBs using Au NPs in A549 cells and the underlying mechanism. 2. Do you consider the topic original or relevant in the field? Does itaddress a specific gap in the field? Since Au NPs have been widely explored as biomedicine, cytotoxicity of them is not a new topic (for instance Pan, Yu, et al. "Size‐dependent cytotoxicity of gold nanoparticles." Small 3.11 (2007): 1941-1949.) However, authors were able to systematically investigate the toxicity with the special focus on DUB enzymes in A549 cells and provided valid mechanism discussion on how the cell viability is decreased via reactive oxygen species generation. 3. What does it add to the subject area compared with other published
material? See above. 4. What specific improvements should the authors consider regarding the
methodology? What further controls should be considered? I have no recommendation for improvement. The experiment design workflow is valid, and the overall quality of figures and manuscript is well prepared. 5. Are the conclusions consistent with the evidence and arguments presented
and do they address the main question posed? Yes. 6. Are the references appropriate? Yes. 7. Please include any additional comments on the tables and figures. In general figure X and Y axis label font size could be unified to improve the presentation.
Reviewer 3 Report
I suggest the authors should better discuss the choice of A549 cells, and also the choice of inhibitors used in the experiments.
Also, the discussion would gain of a broader analysis of the literature concerning other types of NPs than AuNPs, acting on the ubiquitin proteasome system.
English language editing is critically needed.
Fig 5A : Alexa FLUOR and not Flour
